# The Sublingua of *Lemur catta* and *Varecia variegata*: Only a Cleaning Function?

**DOI:** 10.3390/ani15020275

**Published:** 2025-01-20

**Authors:** Juan Francisco Pastor, Magdalena Natalia Muchlinski, Josep María Potau, Aroa Casado, Yolanda García-Mesa, José Antonio Vega, Roberto Cabo

**Affiliations:** 1Departamento de Anatomía y Radiología, GIR “Osteología y Anatomía Comparada”, Universidad de Valladolid, 47005 Valladolid, Spain; juanpas@uva.es (J.F.P.); 2Anatomical Sciences & Education Center, Oregon Health and Science University, Portland, OR 97239, USA; muchlins@ohsu.edu; 3Unidad de Anatomía y Embriología, Universidad de Barcelona, 08036 Barcelona, Spain; jpotau@ub.edu (J.M.P.); aroa.casado@ub.edu (A.C.); 4Department of Evolutionary Biology, Ecology and Environmental Sciences, University of Barcelona, 08028 Barcelona, Spain; 5Departamento de Morfología y Biología Celular, Grupo SINPOS, Universidad de Oviedo, 33006 Oviedo, Spain; garciamyolanda@uniovi.es (Y.G.-M.); javega@uniovi.es (J.A.V.); 6Instituto de Investigación Sanitaria del Principado de Asturias, ISPA, 33011 Oviedo, Spain; 7Facultad de Ciencias de la Salud, Universidad Autónoma de Chile, 7500912 Providencia, Chile; 8Departamento de Anatomía e Histología Humanas, Universidad de Salamanca, 37007 Salamanca, Spain

**Keywords:** *Lemur catta*, *Varecia variegata*, lemur, sublingua, scanning electron microscopy

## Abstract

The sublingua is an anatomical structure located under the tongue. In addition, prosimians have modified lower incisors and canines called the “dental comb”, and classical authors assign the sublingua the role of a toothbrush. Comparative studies of macro and microanatomy are scarce or incomplete for primates. Thus, the putative function of this unique feature remains unclear. To better understand the functional significance of the sublingua, we studied this structure in *Lemur catta* and *Varecia variegata* specimens using histochemical staining techniques together with scanning electron microscopy with microanalysis. The new data obtained provide a fuller picture of the role assigned to sublingua so far in some prosimians, which seems that it could be more complex, in light of the morphological findings, than just dental hygiene.

## 1. Introduction

The sublingua is a rare sublingual structure of the oral cavity of some animals [1]. As its name implies, it is located below the tongue and is present and fully developed in some primate species (Prosimians: lemurs (Lemuridae) and loris (Lorisidae)) but can be seen in its more rudimentary form in marsupials (Diprotodontia), Tupaia (Scandentia) and colugos (Dermoptera). In addition to the sublingua, Prosimians have modified incisors and canines in the lower jaw called a “dental comb” [2]. The incisors and canines of the dental comb are placed in a horizontal plane with the apex of the teeth oriented cranially. The sublingua of lemurs have an arrowhead appearance, are flattened from dorsal to ventral, and decrease in width from caudal to cranial, finishing in a cranial tip which is usually keratinized. Located immediately below the tongue, it is covered by mucosa and the ventral surface is keratinized in adults. The morphology and location of the sublingua led some authors to conclude that it functioned as a second tongue. The dorsocaudal part is attached to the tongue by a medial attachment, similar to a frenum, and the cranial part remains free. The free tip varies between species and can present in different shapes, such as a single tip, a tip with several projections or a wider and rounded tip. This varied morphology may be attributed to differences in feeding behavior [3]. The lateral edges can be smooth (e.g., *Varecia variegata*), slightly serrated (e.g., *Lemur catta*), with various projections of variable size (e.g., *Microcebus murinus*) or can have finger-like keratinized processes like in *Tupaia belangeri* [4].

The sublingua is housed in the oral cavity, so we first must consider what the oral cavity means in a primate. The oral cavity is central to somatosensory environment exploration (mainly to feed), but it also serves an important social component (i.e., social grooming and vocalization), but fundamentally, it serves as an important first step in feeding—digestion via secretions. The oral cavity is also the first line of defense with regard to immunity. The function of the sublingua in lemurs may be related to teeth cleaning, especially for those teeth affiliated with the dental comb [1]. Pocock [5] says that the sublingua “is a purely functional organ, evolved from some part of the tongue for the purpose of cleaning the functionally specialized lower incisor teeth”. Lemurs have no free claws on the fingers and only one specialized on the toes, so they use the “dental comb” for cleaning the woolly hair in grooming sessions between individuals in a colony [6]. The teeth would be cleaned by sliding the tip of the sublingua back and forth, passing the projections of the tip between the teeth.

Since the 19th century, several researchers have provided outstanding macroscopic studies of the sublingua [5,6,7,8,9,10]. At the end of the 20th century, other authors such as Hofer [4,11,12,13,14,15,16] and Rommel [1] provided updated knowledge of its structure. However, most studies about some primate species, published or presented at conferences, have focused on behavioral analysis, while morphological studies are less numerous. Comparative studies of macro and microanatomy in Lemurs are scarce and incomplete, causing a deficit of morphological data to interpret the possible functions of the structures analyzed. In our research, we had the opportunity to study the sublingua of two lemur species: *Lemur catta* and *Varecia variegata*. Previously, the sublingua was described in *Lemur catta* but mainly using macroscopic techniques like low magnification optic microscopy [3,5,6,10,14], and, to our knowledge, the sublingua in *Varecia variegata* has never been described in detail. Here we used structural techniques, semi-thin sections and scanning electron microscopy with microanalysis to study the morphology and structure of these two lemurs in detail. The study aimed to provide further information about this unique organ and its possible functional significance.

## 2. Materials and Methods

### 2.1. Specimen Acquisition and Preparation

The study was conducted according to the guidelines of the Declaration of Helsinki and approved by the Institutional Review Board (or Ethics Committee) of the University of Barcelona (protocol code 00003099, 18 September 2017).

The sublingua were removed from six *Lemur catta* (two females and four males) and five *Varecia variegata* (two females and three males). The primates used in this study were not euthanized for this study but rather obtained opportunistically from Spanish zoos (Zoo Santillana del Mar, province of Cantabria; Bioparc Fuengirola, province of Malaga; Bioparc Valencia, province of Valencia) when animals died due to natural causes. It is important to note that these animals were fed special commercial diets for lemurs supplemented with fresh fruits.

Whole body specimens were freshly frozen at the source (zoo) and then later shipped in coolers to preserve the cold chain to the Anatomy and Radiology Department (Anatomical Museum collection, Universidad de Valladolid). Neither oral/dental pathology was previously reported by the donors of the specimens, nor was it observed while the samples were taken. To obtain the tongue and sublingua, the animals were allowed to thaw for 24 h in a refrigerator. The jaw was lowered, and the tongue was pulled out with forceps. Once the tongue had been placed in antepulsion, it was sectioned at the level of the glossoepiglottic vallecula using a scalpel. The samples obtained were immersed in 10% formaldehyde in a glass jar marked with species, sex and date of collection and kept until their use. After the tongue and sublingua were removed, the remainder of the specimen was refrozen for further investigation. It was not possible to obtain the age of the specimens due to the origin of the animals, as many of them had been donated years before proposing the study. The sequence of ages was estimated, assuming that the size and some morphological features could give an idea of the approximate sequence of ages of the specimens. We estimated the age of the specimens according to the body size and length, the degree of keratinization and pigmentation and the size of the tongue and sublingua. Regarding *Lemur catta*, the ages were classified into three stages (sub-adult, adult and elder). In the case of *Varecia variegata*, two stages were established (sub-adult, adult).

### 2.2. Macroscopic Observation of Samples

Together with direct observation, was used a Leica M205FA semi-automatic stereo microscope (Leica, Wetzlar, Germany) for transmission (contrast and relief), reflection and fluorescence studies. Optics: 1X (working distance 61.5 mm), 2× (working distance 20 mm) and 5× (working distance 19 mm), with a zoom from 0.7 to 20.5×. An LED ring provided the reflection light with 1/4, 1/2 or full-ring illumination. The digital color camera used in this study, Leica DFC310FX, had a maximum resolution of 1392 × 1040 pixels (1.4 Mpixels CCD). The excess fixative liquid was removed to avoid reflections, and the tongues were fixed with needles to a cork base to avoid sublingual movements. Then the samples were observed, and some photographs were taken from the ventral aspect of the sublingua. The equipment belonged to the Scientific Technical Services of the Universidad de Oviedo. Although the specimens are primates, the terms used for the descriptions were those used in veterinary terminology as they are not fully bipedal animals. The terms “cranial/caudal” were used as synonyms for “superior/inferior” and “dorsal/ventral” were used as synonyms for “posterior/anterior”.

### 2.3. Fixation and Processing for Histological Examination

The sublingua was fixed by immersion with 10% formaldehyde in 0.1 M phosphate buffer (room temperature, pH 7.2) for 24–48 h. Then, each sublingua was divided into several samples through cuts perpendicular to the longitudinal axis, except for three sublingua of each species that were kept for scanning electron microscopy, as described above (see Materials and Methods Section 2.4). Thereafter, some samples were processed for Masson’s trichrome staining method: dehydrated and embedded in paraffin, cut into 10 µm thick sections perpendicular to the sublingua longitudinal axis, stained (following the instructions of Masson’s trichrome staining kit, Poly Scientific R&D Corp., New York, NY, USA) and finally mounted on gelatin-coated microscope slides. On the other hand, selected samples of the sublingua were post-fixed in 2% glutaraldehyde in 0.1 M phosphate buffer, embedded in Durcupan ACM resin using a commercially available kit (Fluka, Chemie AG, Buchs, Switzerland) following the instructions of the kit. The resin-embedded tissues obtained were cut with a Reichert Jung Ultracut E ultramicrotome (Reichert, Wetzlar, Germany) into 1 µm thick sections and stained with toluidine blue (one minute at 60 °C), then rinsed in distilled water and finally were mounted and examined. Sections obtained by both staining methods were photographed on a Nikon Eclipse 80i optical microscope (Nikon, Melville, NY, USA) coupled to a Nokia DS-5M camera (Nokia, Espoo, Finland).

### 2.4. Scanning Electron Microscopy with Microanalysis

The study was restricted to the ventral surface of the sublingua (3 specimens of each species) because it is the surface that faces the “dental comb”. The samples were immersed in a 1 N solution of hydrochloric acid concentrated at room temperature, ranging from 10 to 20 min (depending on the sample size). This treatment was performed to remove debris from sloughed cells. Then, after rinsing several times in tap water, the pieces were dehydrated in a series of acetones of increasing concentration, critical point dried in a Balzers CPD 030, sputter coated with 3 nm gold in a Balzers BAL-TEC SCD 050 and examined under a conventional scanning electron microscope (MEB JEOL-6100, JEOL, Tokyo, Japan) with a tungsten filament electron cannon, 0.3 to 30 kV, equipped with secondary electron detector and backscattered electron detector. As allowed by the equipment, an analysis of the composition of elements based on energy dispersion X-ray spectroscopy (EDX) assisted with INCA Energy 200 software was performed to detect the composition of elements except those of very low weight (B, Be, Li, He, H).

## 3. Results

### 3.1. The “Dental Comb”

The mandible was examined in both species studied, *Lemur catta* (Figure 1a) and *Varecia variegata* (Figure 1c), focusing on the area against which the sublingua (Figure 1b,d and Figure 2a–d) is suggested to perform a cleaning function. Both species have nearly identical dental combs and there were no sex or species-specific differences observed concerning the mandible morphology (Figure 3a–d’) in contrast with those found when analyzing the sublingua (as described in results Section 3.2).

### 3.2. Lemur catta Sublingua

Macroscopic observation showed the typical “arrowhead or sheet-shaped” structure. The ventral (inferior) surface has longitudinal folds (rod-shaped or ridge-shaped, Figure 1b, Figure 2a,d, Figure 4a, Figure 5a–g’ and Figure 6a). In *Lemur catta*, this arrowhead-shaped organ ends in a narrow tip and has slightly serrated edges on the lateral margins (Figure 5 and Figure 6). The males have a tip with a longitudinal rod that does not reach the cranial limit (Figure 5a–e,a’–e’,g,g’), while rods from the females studied do reach the cranial limit (Figure 5f,f’). The central rod reaching the tip results in a trifurcated tip in females, while males have a single tip (Figure 5). Such a comparison could not be made in *Varecia variegata* because one of the two female specimens studied had a damaged and incomplete tip. The rest of the organ presents three longitudinal folds (Figure 2, Figure 4 and Figure 5) on its ventral surface (a central rod with two parallel adjacent ridges on each side). A previously unobserved feature of the sublingua’s inner structure was reported using toluidine blue semi-thin sections for the first time. Figure 7c,d shows this feature consisting of structures located in the epidermal layer, very close to the ventral surface that could be compatible with the section of a duct.

Using scanning electron microscopy (SEM), different papilla-like structures on the ventral surface of the sublingua were observed in some specimens of *Lemur catta*. The elder specimens presented a sublingua with a smooth ventral surface without the aforementioned structures. Animals classified as adults had a smooth cranial part, also without such structures (Figure 6h,i), in contrast with a rough caudal part (Figure 6i–l) with mechanical papilla-like formations. The sub-adult specimen had a roughened surface and papillae formations of different shapes along the entire ventral surface.

The SEM images confirmed the typical keratinized surface with pores. Some of these porous structures were covered with an amorphous substance (Figure 7b). SEM microanalyses revealed iron as a component of the amorphous substance (Figure 7b), while the rest of the surface lacked these pores and the substance (Figure 6 and Figure 7a).

### 3.3. Varecia variegata Sublingua

Macroscopic observation showed the typical triangular arrowhead shape, but with a wider tip presenting several extensions or denticles along the lateral parts, making it in some cases similar to the tip of a brush (Figure 1d, Figure 2c,d, Figure 8a and Figure 9b). The optical microscopy analysis of Masson’s trichrome-stained sections showed an internal side full of structures similar to cavities or ducts. We can observe zones where such cavities seem to drain partially into the keratinized layer (Figure 8c,d and Figure 10c,d). The presence of these structures is not regular along all sublingua (displayed in histological sections from cranial to caudal, Figure 8b–e). Cranially (Figure 8b), several cavities or ducts in the sublingua could be observed. These cavities or ducts seem to merge and drain to larger cavities towards the caudal part of the sublingua. Caudally, these cavities drain into a chamber or deposit placed between the tongue and the sublingua (Figure 8c–e). The caudal part also presented such spaces, but only partially together with the presence of connective tissue (Figure 5e).

The SEM images of the ventral aspect of the sublingua show the typical keratinized surface, where bright and dark areas can be distinguished (Figure 9a,d,f vs. Figure 9c,e,g). Both the bright and dark areas presented a porous surface (Figure 9d,f vs. Figure 9e,g), but only the dark areas were filled with an amorphous substance (Figure 9c,e,g). The SEM microanalysis reveals that chlorine is the differential substance found within the pores of the dark areas (Figure 10b).

## 4. Discussion

For over 150 years, the sublingua has been purported to be the dental comb-cleaning organ among the lemurids [5,6,7,17] because of its highly keratinized tip, its placement under the tongue and its direct contact with the modified teeth of the lower jaw. FW Jones [6] hypothesized that the dental comb in colugos and lemurs is an adaptation to clean and groom the woolly hair of these animals, because it is difficult to clean through other methods like scratching. Jones [6] adds the reflection that the sublingua “is a purely functional organ, evolved from some part of the tongue for the purpose of cleaning the functionally specialized lower incisor teeth”. However, Ankel-Simons [18] writes, “ever since Bluntschli [17] first introduced the functional interpretation of the sublingua in prosimians as a toothbrush, there has been little or no clear confirmation of this function. Even now, we lack careful anatomical and functional studies of the sublingual organs of most primates”. We totally agree with this reflection. With the various microscopic approaches outlined in this study, we now have evidence to expand the putative functional roles the sublingua serves. We suspect it serves as a cleaning tool based on our microscopic observations. However, the species-specific diversity in sublingua anatomy documented in this study could suggest that it may serve additional functions (e.g., foraging, digestion, scent marking) as we will discuss below, although further studies are needed to confirm this.

To establish the putative alternative roles for the sublingua, we also must take into account, in addition to its morphology, the oral structures surrounding it and the oral cavity’s several roles. The oral cavity is critical for digestion, sound articulation and immunity, and it also serves as a sensory structure that can also aid in object manipulation/exploration. For instance, the mucosa distributed along the palate, mouth floor, gingiva and vestibular part of cheeks and lips perform digestive, secretory and sensory roles. Moreover, the tongue is a perfect example of a multipurpose organ that is functionally complex and serves more than just a chemoreceptive role. The tongue provides information to the brain about food’s nutritional properties, assists with modifying vocalizations and can serve as a valuable tool for assessing and manipulating objects [19]. Therefore, we think the so-called “second tongue”, the sublingua, could also be multifunctional. Below, we highlight the anatomical and physiological evidence that supports our multifunctional hypothesis.

### 4.1. Non-Gustatory Mechanical Papillae

In *Lemur catta* specimens there appears to be an age-related pattern in the distribution of mechanical papillae over the ventral surface of the sublingua. In the smallest specimen, we observed various structures resembling mechanical papillae throughout the ventral surface of the sublingua. One specimen, classified as intermediate in size, had these mechanical papillae on the caudal part of the ventral surface. In the largest specimen, which is assumed to be the oldest, we found a smooth surface without mechanical papillae. Assuming the relationship between size and age is accurate, these findings could indicate that the rough non-gustatory papillae disappear in a pattern from cranial to caudal areas as the animal ages. Age-related position, density and keratinization changes have been observed among other primates [14,16,20,21]. Pastor and colleagues [21] claim that the free portion of the tongue (which is longer in adults) evolves from birth to adulthood as an animal tongue’s functional roles develop through time. Hofer [13] also found structures described as papillae on the sublingua’s lateral folds in *Microcebus murinus,* forming a “comb” of papillae, which remains functionally enigmatic. These non-gustatory mechanical papillae observed in our sample could fulfill the previously proposed cleaning function of the dental comb, or it is possible that these papillae can help to disperse some substances. For example, we found an iron-containing substance in the micropores of *Lemur catta*. The iron substance could serve a digestive purpose or some other currently unknown purpose that could be dependent on habitat and/or an animal’s unique life stage. We hypothesize that these mechanical papillae could spread a substance (e.g., iron in *Lemur catta*) over the woolly hair as the hair passes over the ventral surface of the sublingua and dental comb. The substance could change the fur’s odor or trigger other social behaviors, given the weight that olfaction plays in communication in these animals [22].

### 4.2. Sublingua Tip Morphology

The morphology of the sublingua’s tip was shown to vary between species with different dietary preferences [3]. *Varecia variegata*, a non-destructive flower feeder (nectar), has a feathered and highly bifurcated sublingua’s tip, while *Lemur catta* lacks a feathered tongue. Here, we discuss the sex-based differences observed in the *Lemur catta* sublingua’s tip. In the female specimens, we observed highly keratinized denticle-like expansions in both female specimens studied, which were more developed in the larger specimen. Hofer [13], examining the sublingua in *Microcebus murinus*, reports a few tiny orally pointing processes at the two lobes of the sublingua tip in females. Hofer [13] focused on adult females because they have the best-developed highly cornified denticle-like process between the anterior lobes of the sublingua. To confirm our observations, this sex-based difference should be further studied in additional specimens and species.

In addition to denticle-like expansions, there are other sex-based differences among the sublingua of *Lemur catta* studied. The sublingua has a central rod (or keel), which does not reach the tip sublingua in males but does in females. This keel expansion results in a trifurcated tip (a central rod plus the two denticles-like expansions) in females, while the males present a single “arrowhead-shaped” tip. The morphological differences of the tip in both sexes provide some evidence that the variation we are observing may serve more than just a toothbrush function. The same functional role would not demand differences in the sublingua’s morphology because the lower jaws do not present significant gender or species-specific differences.

### 4.3. Central Cavities in Varecia’s Sublingua

The inner structure of *Varecia variegata* sublingua is quite different from *Lemur catta*. In *Varecia variegata*, we saw a set of cavities that converge towards the dorsocaudal part of the sublingua, where it merges with the tongue and ends in a unique space between the tongue and the sublingua. We think these spaces could be filled with fluid to lengthen the sublingua, as occurs in the cavernous bodies of the penis, as *Varecia variegata* nectar-feed in flowers with deep nectaries. In a previous study comparing the tongues of three lemur species [3], *Varecia variegata* was found to have a wavey and folded tongue in its relaxed position, suggesting that the tongue in this species could be greatly elongated. Additionally, we show small projections that appear to move from these central cavities towards the ventral keratinized layer in which we identified dark areas that appear to be secreting chlorine pores, as opposed to the bright regions where no substances were detected by the backscattered electron technique. We could not establish if the projections towards the keratinized layer were related to the surface porous associated with the presence of chlorine; further studies are required. In addition, we also saw in one *Varecia variegata* specimen pseudopapillae crossed with a central canal, present in the most posterior part of the sublingua ventral surface (unpublish data). Moreover, in *Lemur catta*, we found porous areas on the keratinized surface, some of which were partially filled by an amorphous substance. Iron was the differential component of the substance, and was associated with the porous areas of the ventral keratinized surface. We think the porous areas on the keratinized surface in both species are functional and the pores are secretory. The fact that these proposed secretory structures are not present on the rest of the sublingua suggests that sublingua may serve more than just one function (i.e., dental cleaning) but rather may have implications for digestion [23] or scent marking. Since the morphology and arrangement of the dental comb are very similar in both *Varecia* and *Lemur* (nearly identical), there should be many similarities in the morphology of the sublingua between the two species if the only purpose of the sublingua was to serve as a toothbrush, and our data showed key differences.

## 5. Conclusions

Based on the evidence presented in this study, we cannot support the single-function hypothesis for the sublingua. The documented intra and interspecific differences, combined with the presence of chemical substances within micropores on the ventral surface of the tongue, suggest that we should consider additional roles/functions of the sublingua, including but not limited to food processing, grooming or social behavior.

Comparative studies of macro and microanatomy are scarce or incomplete for these primates, and although there are some impressive and comprehensive studies on many primate species, we need targeted behavioral studies to better understand the possible functional roles of the sublingua. The sublingua of the *Lemur catta* has been better studied, while very little is known about the sublingua of *Varecia*. Here, we add new morphological data about *Lemur catta* and present for the first time images from the *Varecia variegata* sublingua ultrastructure that could aid in a better understanding of the organ. Despite the limitations of this study, such as (1) no age data, (2) fresh frozen and thawed specimens vs. quick perfusion fixation and (3) sample size, the novel and detailed macro and microanatomy presented here are valuable—not only to confirm the past studies, but also to help shape future research towards better understanding the ecology, behavior and anatomy of these endangered primates.

## Figures and Tables

**Figure 1 animals-15-00275-f001:**
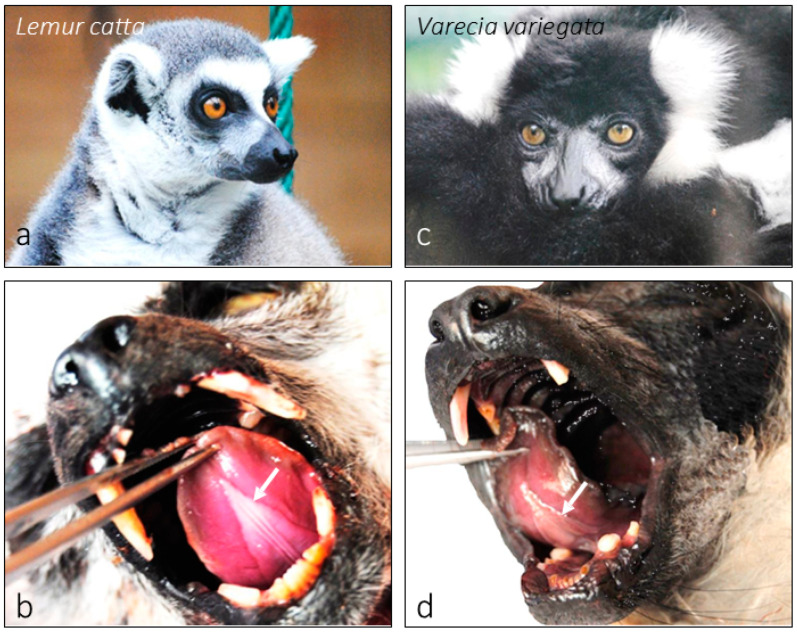
Images of *Lemur catta* (**a**,**b**) and *Varecia variegata* (**c**,**d**). Images (**a**,**c**) were taken from specimens hosted in the Santillana del Mar Zoo. Images (**b**,**d**) show the position of the sublingua in the oral cavity under the tongue.

**Figure 2 animals-15-00275-f002:**
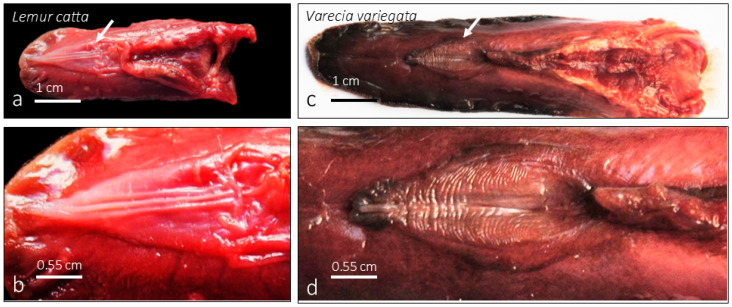
The ventral aspect of the unfixed tongue and sublingua of *Lemur catta* (**a**,**b**) and *Varecia variegata* (**c**,**d**) shows the comparative size of such structures. The surface of the *Varecia variegata* sublingua presents several transversal folds similar to the folds that were previously described on the tongue’s dorsal surface by Pastor et al. [3], which was related to the way of feeding.

**Figure 3 animals-15-00275-f003:**
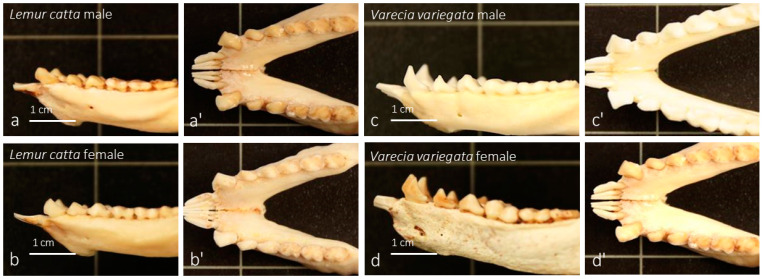
The images (*Lemur catta* on the left and *Varecia variegata* on the right) display the lower jaw or mandibular bone with the so-called “dental comb” (modified lower incisors and canines), the surface against which the sublingua would act as a toothbrush according to several classical studies. The images placed to the right of the images identified with the specie and gender correspond to the dorsal view of the same sample.

**Figure 4 animals-15-00275-f004:**
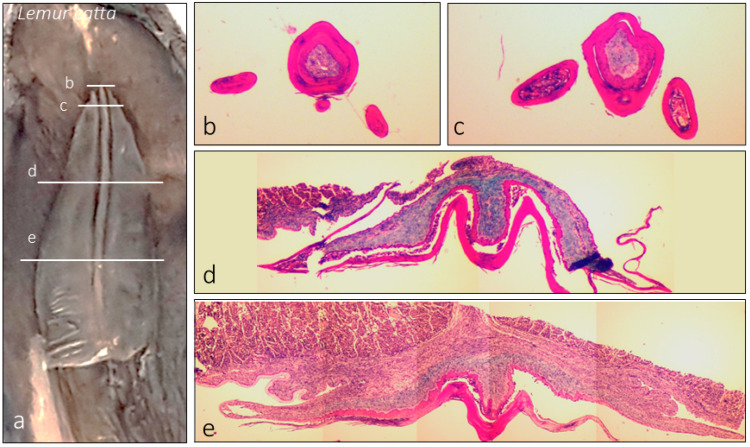
Images of the internal structure of the sublingua of *Lemur catta* stained with Masson’s trichrome. On the left (**a**), we summarize the approximate plane of the transversal cross-section shown on the right (**b**–**e**).

**Figure 5 animals-15-00275-f005:**
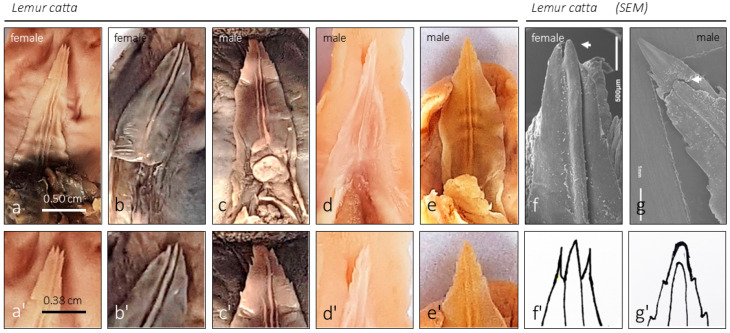
Ventral aspect of the sublingua of several *Lemur catta* specimens ((**a**–**e**) and slightly enlarged in the second row of images (**a’**–**e’**)). The sublingua’s tip could be observed. SEM images of a female (**f**) and male (**g**) specimen magnify the morphology of the tip. The drawings presented in (**f’**) (female) and (**g’**) (male) summarize the differences found between genders (3 females versus 4 males) regarding the morphology of the tip. In females, the central rod reaches the most cranial part of the tip, which is not the case in males. Such a comparison cannot be made in *Varecia variegata* because one of the 2 female specimens studied has a damaged and incomplete tip.

**Figure 6 animals-15-00275-f006:**
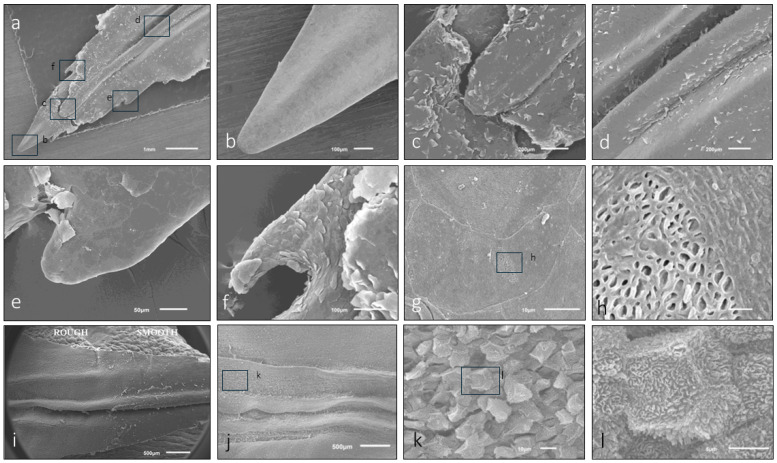
SEM images of *Lemur catta* sublingua’s ventral surface. The tip of a male specimen with some areas magnified (**a**–**f**). In the “(**i**)” low magnification SEM image, we could appreciate an anterior smooth surface (left side) and a posterior rough surface (right side). Images (**g**,**h**) show the ventral smooth anterior surface with a keratinized area with some porous zones (magnified in (**h**)) presented sparsely along such surfaces. With different magnifications (images (**j**–**l**)), the morphology of the rough posterior surface caused by several pseudo papillae, similar to mechanical ones, could be appreciated. Scale bar image (**h**), 1 μm.

**Figure 7 animals-15-00275-f007:**
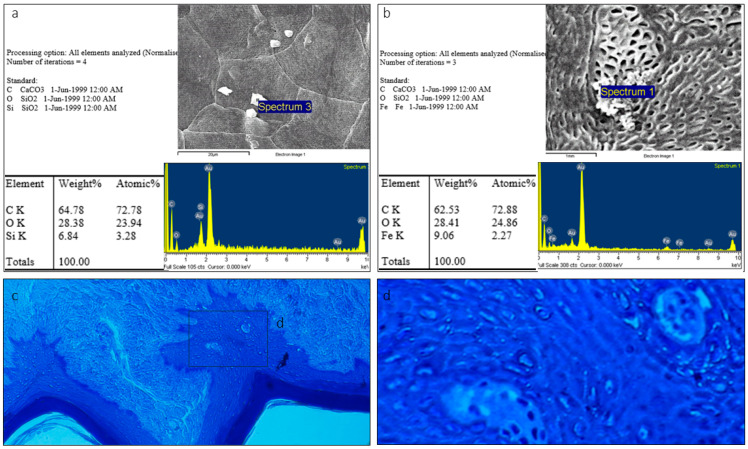
Analysis of element composition based on backscattered electrons with the point of measurement (spectrum) in *Lemur catta* samples. We focused on keratinized areas with and without porous surfaces, described in Figure 6g,h. On keratinized and nonporous surfaces, the analysis yielded the presence of carbon and oxygen (not shown in the figure). In contrast, we also found iron in the composition of the substance in the porous surface (image (**b**)). On the non-porous keratinized surface, we found a crystallized substance, which through analysis was determined to be silica in composition (image (**a**)), making it compatible with the grains of sand found in the animal’s habitat. Toluidine blue-stained transverse sections focused on the keratinized layer of *Lemur catta* sublingua with non-keratinized zones (**c**,**d**).

**Figure 8 animals-15-00275-f008:**
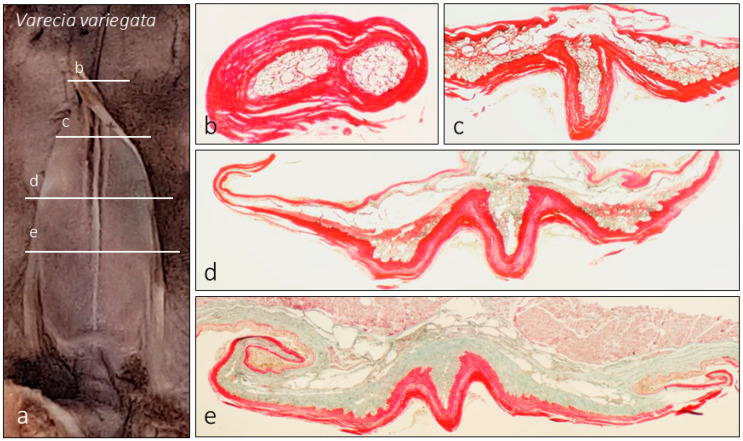
Images of the internal structure of the sublingua of *Varecia variegata* stained with Masson’s trichrome. On the left (**a**), we summarize the approximate plane of the transversal cross-section shown on the right (**b**–**e**). Note that the inner structure is filled with cavities (**b**–**e**).

**Figure 9 animals-15-00275-f009:**
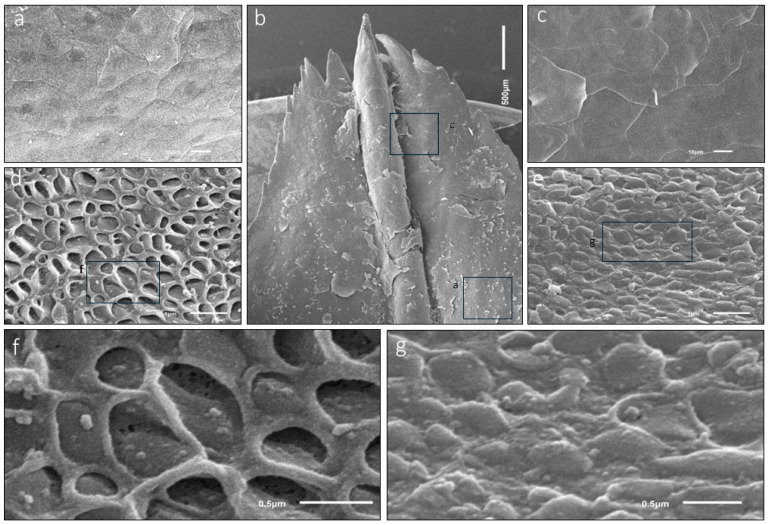
Different images were obtained using the SEM technique from the ventral surface of the sublingua in *Varecia variegata*. In image (**b**), we displayed the zone studied. We found bright (**a**,**d**,**f**) and dark (**c**,**e**,**g**) areas. We could observe a porous surface in bright areas without any substance filling them. In contrast, we show that the porous surface is filled with a substance in the dark areas. The substance was analyzed, and the results are displayed in Figure 9.

**Figure 10 animals-15-00275-f010:**
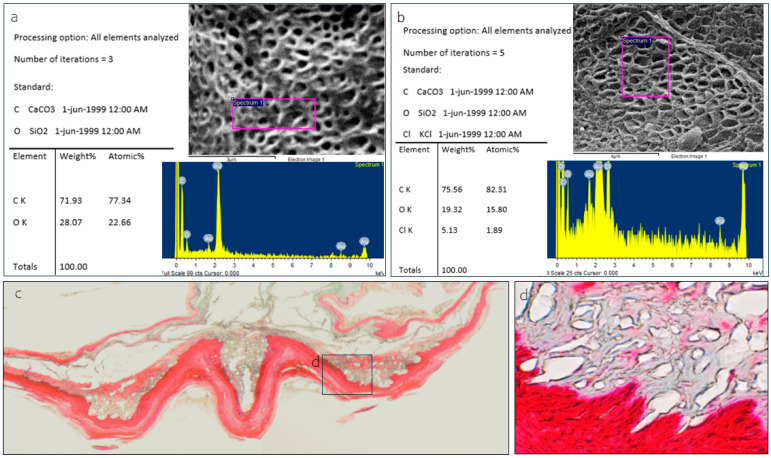
Analysis of element composition based on the backscattered electrons technique and the point of measurement (spectrum) in *Varecia variegata* sublingua samples, focusing on the bright and dark areas described in Figure 9. The analysis of a porous region without any filling substance yielded only carbon and oxygen (**a**), whereas the porous region filled with a substance also yielded chlorine in the composition (**b**). Masson’s trichrome-stained sections of a *Varecia variegata* sublingua transverse section show cavernous-like structures with projections towards the keratinized layer (**c**,**d**).

## Data Availability

The original contributions presented in this study are included in the article. Further inquiries can be directed to the corresponding author(s).

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
