# Peer review of "The Sublingua of Lemur catta and Varecia variegata: Only a Cleaning Function?"

_animals, 2025, doi:10.3390/ani15020275_

Round 1

Reviewer 1 Report

Comments and Suggestions for Authors

As indicated in the attached file

Methods of gross observation has to be explained

Whether H & E staining was carried out or not but it was explained in results

In the results figure  number should be mentioned properly like 6a, 6b etc

Figure 5c was not mentioned in the results

Figures should be mentioned only in the results not in discussion

References in the discussion has to be modified according to MDPI format 

Comments on the Quality of English Language

English language has to be improved and the grammatical errors has to be rectified. Hence the article require major revision.  

Author Response

Thank you very much for your extensive revision. I have remarked with light green color the parts of the text modified following your advice and indications that detect many errors, most of them grammatical errors. 

1. Methods of gross observation has to be explained

Response: I have added more information to the material and methods and rewritten some parts (all remarked with light green color). 

2. Whether H & E staining was carried out or not but it was explained in results

Response: that fact is an error and was corrected and suppressed (we used the material and method of the paper studying the tongue because some of the studies were made at the same time and with almost the same techniques).

3. In the results figure  number should be mentioned properly like 6a, 6b etc

Response: The figures were detailed as the revision indicates.

4. Figure 5c was not mentioned in the results, figures should be mentioned only in the results not in discussion

Response: Figure 5c was cited in line 205 and 210, all the mentions to figures in discussion sections were removed. 

5. References in the discussion has to be modified according to MDPI format

Response: References were adapted to the MDPI format. We only cited the name of the authors in case we wrote a literal textual quote.

6. During freezing ice crystals  will form and it will alter the morphological features. whether the author compared the samples with control 

Response: About the way of preserving the specimens we receive donations preserved by freeze and we know the limitations of the study, facts that were remarked in the conclusions, but we can not do anything more than keep the freezing chain. We did not observe artifacts caused by the method and we have experience of working with such kinds of samples. If the freezing method is correct the samples preserve good. All samples were subjected to the same process, no samples were subjected to treatment, leaving no controls. In transmission electron microscopy and to see the most delicate details inside the cells, ultra-rapid freezing is chosen as the method that best preserves. Our freezing has not been like this, but it has been sufficient to not cause damage to the tissue that would distort the results, as we have observed when analyzing the samples.

7. Normally sections were cut at 3 to 5 micron what is the use of 10 micron thickness 

Response: We also used it because we had been working for more than 25 years with sections between 8 and 12 micrometers and when the samples are well preserved, we can get very well images as other reviewers told us. For Toluidine blue we use to cut between 0,5 and 1 micrometers. We also improved the explanations of parts that, as your revision indicates, could be confusing.

8. English language has to be improved and the grammatical errors have to be rectified. 

Response: The English language was revised by professional software assisted by an English language translator and the grammatical errors (gently indicated by the reviewer and others) were rectified.

9 Comments about the footnote of figure2.

Response: It was described in pastor et al 2021 as it is indicated in this paragraph but we replaced the reference for more clarity.

10. About the attached pdf with comments.

Response: We asked for those comments in the same text (that we attach with the name “reviewer 1 animals-3375714-answered” pdf) and in the main text were remarked with light green color.

Reviewer 2 Report

Comments and Suggestions for Authors

See attached PDF.

Comments on the Quality of English Language

See attached PDF.

Author Response

Thank you very much for your revision and advice. I added to the abstract the text suggested. I remark the changes made in yellow. I explain the similarities in mandibular bones and indicate the differences in the sublingua morphology.

General Comments: This is an interesting well researched project that I feel should be in the literature. This is very solid straight forward functional morphological research that uses detailed visualization to help develop hypotheses about the adaptive function of the sublingua in lemurs. I think all sections are sound and I appreciated the development at the end of workable hypotheses. The only place where additional work is needed is in editing the English grammar throughout. I have not provided text editing, but I am willing to do so if it is needed.

  1. Abstract: I think it would be worth noting toward the end some of the hypothesized functions you propose in the text. Check throughout the manuscript your use of sublingua vs. sublingual.

Response: That text was added to the abstract (line 33 to 36).

  1. Introduction: Overall this section is sound. This said I think the lack of morphological studies and data is slightly overstated.

Response: We change the redaction of such a statement in lines 74 to 82. However, and especially in the case of Varecia, we have not found any studies on the morphology of the sublingual. We usually attend conferences and meetings on Primatology, and we are almost always the only ones who present detailed morphology works, the vast majority being works on behavior and adaptation to their habitat.

  1. Materials and Methods: This section is good, and I have no substantive changes to suggest beyond the grammatical.

Response: The English language was revised by professional software assisted by an English language translator and the grammatical errors (gently indicated by the reviewer and others) were rectified.

  1. Results: You note at the beginning that few dimorphic or species differences were found yet you outline several such differences in the results section.

Response: We didn´t find differences in mandible morphology but we found differences related to the sublingua inner structure compared lemur catta with Varecia variegata and in the superficial morphology in both Lemur catta  males vs females and even in different ages. We add the following sentence: “both species have nearly identical dental combs and there were no sex or species-specific differences observed concerning the mandible morphology (figure 3a to 3d’) in contrast with those found (described in results 3.2) analyzing the sublinguas.” in lines 168 to 171.

  1. Discussion: I like this section very much. I don’t think you overextend yourself in the development of new functional hypotheses. It might be nice to present previous and proposed hypotheses in a tabular form.

Response: We think that there are many figures in the paper so we decided not to add more figures (we have more) or tables, it could be an interesting option but the other 4 reviewers agree with the discussion.

  1. Figures and Tables: This are all very helpful. I’ll leave it to a trained histologist to provide a more detailed critique; References: Appear thorough and complete.

Response: Thank you for your comment. We try to use as many techniques as possible considering the number of samples we get.

Reviewer 3 Report

Comments and Suggestions for Authors

The authors describe a particular salivary gland in two endagered species of primates. These lemurs, belonging to a group of prosimians, can make an interesting contribution to the understanding of the anatomy of similar glands in more evolved primates. The number of specimens is valid and the procedures folllowed appropiate. Figures and references are good. The results seems convincing that this particular gland can be plays a role not only during chewing but also with other functions such as to clean teeth, mouth, tongue and probably in social beaviours. 

I think that this manuscript deserves to be published in this form. I noted only a small typo error in pag. 11 line 363: please change "variegata" instead of "variegatta". 

Author Response

Thank you very much for your revision and advice. I added the changes suggested and others suggested by the rest of the 5 referees so the text present 5 colors remarking the changes.

The authors describe a particular salivary gland in two endagered species of primates. These lemurs, belonging to a group of prosimians, can make an interesting contribution to the understanding of the anatomy of similar glands in more evolved primates. The number of specimens is valid and the procedures folllowed appropiate. Figures and references are good. The results seems convincing that this particular gland can be plays a role not only during chewing but also with other functions such as to clean teeth, mouth, tongue and probably in social beaviours.

I think that this manuscript deserves to be published in this form. I noted only a small typo error in pag. 11 line 363: please change "variegata" instead of "variegatta".

Response: The mistake was corrected and revised all the similar terms in all the text.

What is the main question addressed by the research?

The aim of this research is to evaluate if the study of the morphology and structure of the sublingua of two lemur species could suggest specific functions. For this reason the study has been articulated both in macroscopic and microscopic examination.

  • Do you consider the topic original or relevant to the field? Does it address a specific gap in the field? Please also explain why this is/ is not the case.

This work appears original because the knowledge of the structure of this organ is still unclear. From this point of view this work fills a gap.

  • What does it add to the subject area compared with other published material?

Published material refer overall in macroscopic observations. The advance in the study with this work is based on microscopic observations by means of histological procedures.

  • Are the conclusions consistent with the evidence and arguments presented and do they address the main question posed?

Yes, I agree with the authors. The sublingual seems to have other functions in additions to that of toothbrush or cleaning function. Probably it plays a role in social behaviour if it has a chemoreceptor function. In order to investigate this possibility, the authors could perform more specific histological procedures, such as those based on silver or gold/silver impregnations (for examples Golgi’s staining, Bielschowsky’s stain or others). These procedures could demonstrate the presence and distribution of nerve fibers. Moreover the Authors could spend more attention to describe histologic pictures obtained. The histologic pictures should be better appreciated with higher magnifications. Their legends could explain which tissues are present. It seems rich in connective tissue but probably many nerve fibers are also present. For this reason it could be very useful to do impregnation stain methods.

Response: These histological studies with silver stains or antibodies will be addressed in the next studies that we continue to do to further investigate the innervation and vascularization of the tissue if the quality of future samples received from donors allows us to do so. We have tried to do immunohistochemistry but since the samples were not perfused, we did not obtain results of sufficient quality. For this reason, we decided not to delay the presentation of the rest of the results that we think should be known as soon as possible by primatologists in case they can relate them to studies on behavior.

Reviewer 4 Report

Comments and Suggestions for Authors

The manuscript "The sublingua of Lemur catta and Varecia variegata; only a cleaning function?" is an interesting study. Information on sublingual is limited, so any new information is valuable. The study is conducted on rare species, so despite the small number of preparations analyzed, the information obtained is extremely valuable. The introduction is well written and provides a proper introduction to the later parts. The material and methods are described in an understandable manner. The results clearly present the data obtained. The discussion section is appropriate and discusses the results in a broader context. The text is enriched with numerous photos, which adds to the attractiveness of the manuscript.

I have only minor comments:

- figures should be cited in the order they appear in the paper.

- results should be written in the impersonal form, not “we examined”

- references should be adapted to the requirements of the journal

Author Response

Thank you very much for your revision and advice. I added the changes suggested (remarked in brown) and others suggested by the rest of the 5 referees so the text presents 5 colors remarking on the changes. We change to impersonal language in results. We try to cite the figures in the order they appear but sometimes we make a comparison and should refer to not followed figures. References were adapted.

The manuscript "The sublingua of Lemur catta and Varecia variegata; only a cleaning function?" is an interesting study. Information on sublingual is limited, so any new information is valuable. The study is conducted on rare species, so despite the small number of preparations analyzed, the information obtained is extremely valuable. The introduction is well written and provides a proper introduction to the later parts. The material and methods are described in an understandable manner. The results clearly present the data obtained. The discussion section is appropriate and discusses the results in a broader context. The text is enriched with numerous photos, which adds to the attractiveness of the manuscript.

I have only minor comments:

  1. figures should be cited in the order they appear in the paper.

Response: We try to cite the figures in order, and we make some changes but, in some cases, as we are making a comparison, we cannot follow the order. 

  1. results should be written in the impersonal form, not “we examined”

Response: We made the changes suggested and the language was changed to impersonal style (lines 166,189,196,200,207,211).

  1. references should be adapted to the requirements of the journal

Response:  The references were adapted to MDPI format.

Reviewer 5 Report

Comments and Suggestions for Authors

The manuscript is very well-written and interesting. It takes a morphofunctional approach to the sublingua in two species: Lemur catta and Varecia variegata. Below are some questions and suggestions aimed at improving the manuscript.

- Line 72: Is it really "sublingual," or should it be read as "sublingua"?

- Do any of the studied animals have a history of dental or oral diseases? It is important for this information to be included in the manuscript to prevent any observed variations from being linked to a pathology.

- Please review the use of positional and directional terms such as "anterior," "posterior," "superior," and "inferior." Although they are primates, it may be more appropriate to use veterinary terminology, as they are not fully bipedal animals. Terms like "cranial," "caudal," "dorsal," and "ventral" might be more accurate.

- Add the references for Figures 1a and 1c.

Sincerely,

Author Response

Thank you very much for your revision and advice. I added the changes suggested and others suggested by the rest of the 5 referees so the text presents 5 colors remarking on the changes. We change the positional terms to those used in veterinary terminology and advise of it in the macroscopical description (all the changes noted in blue). I add the reference for Figure 1, all the pictures taken by our research team.

The manuscript is very well-written and interesting. It takes a morphofunctional approach to the sublingua in two species: Lemur catta and Varecia variegata. Below are some questions and suggestions aimed at improving the manuscript.

  1. Line 72: Is it really "sublingual," or should it be read as "sublingua"?

Response: The correct term is sublingua, we change to the correct word in line 72

  1. Do any of the studied animals have a history of dental or oral diseases? It is important for this information to be included in the manuscript to prevent any observed variations from being linked to a pathology.

Response: We add the statement: “neither oral/dental pathology was previously reported by the donors of the specimens, nor was it observed while the samples were taken” in lines 101 to 103.

  1. Please review the use of positional and directional terms such as "anterior," "posterior," "superior," and "inferior." Although they are primates, it may be more appropriate to use veterinary terminology, as they are not fully bipedal animals. Terms like "cranial," "caudal," "dorsal," and "ventral" might be more accurate.

Response: We change the positional terms to those used in veterinary terminology and advise of it in the macroscopical description. Terms like "cranial/caudal” as “superior/inferior" synonym and "dorsal/ventral” as “posterior/anterior" synonym.” in lines 128 to 131 and made that change in lines 177,178,191,192,206,207,247,318,322 and 361.

  1. Add the references for Figures 1a and 1c.

Response: References to figures 1a and 1b were included in lines 166-167 and 222-223. All the pictures were taken by our research team.

Round 2

Reviewer 1 Report

Comments and Suggestions for Authors

Authors carried out most of the suggestions however few corrections indicated in the text has to be carried out

Author Response

Thank you again for the detailed review of the manuscript. The following changes (remarked with yellow color in the text) have been made:

Comment 1: The purpose of 10 micron thickness is not explained (lines 138-139)

Response 1: The text was replaced by "Thereafter, some samples were processed for Masson´s trichrome staining method: dehydrated and embedded in paraffin, cut into 10 µm thick sections perpendicular to the sublingua longitudinal axis, stained (following the instructions of Masson's trichrome staining kit, Poly Scientific R&D Corp, NY, USA) and finally mounted on gelatin-coated microscope slides." explainning the purpose of the 10 micron sections (lines 137 to 141)

Comment 2: line 221, eliminate the text "the species studied"

Response 2: the text was eliminated from line 221 (now line 220).

Comment 3: line 346; figures should be mentioned only in the results not in discussion

Response 3: the term "figure 6" was eliminated from line 346 (now discussion section, line 345).

Comment 4: wrong expression line 403

Response 4: the wrong expresion "not only to by confirming past studies" was sustituted by the correct one "not only to confirm the past studies" (now line 402).